# How Do the Psychological Functions of Eating Disorder Behaviours Compare with Self-Harm? A Systematic Qualitative Evidence Synthesis

**DOI:** 10.3390/healthcare13151914

**Published:** 2025-08-05

**Authors:** Faye Ambler, Andrew J. Hill, Thomas A. Willis, Benjamin Gregory, Samia Mujahid, Daniel Romeu, Cathy Brennan

**Affiliations:** Institute of Health Sciences, University of Leeds, Leeds LS2 9NL, UK; a.j.hill@leeds.ac.uk (A.J.H.); t.a.willis@leeds.ac.uk (T.A.W.); umbdg@leeds.ac.uk (B.G.); d.j.romeu@leeds.ac.uk (D.R.); c.a.brennan@leeds.ac.uk (C.B.)

**Keywords:** eating disorders, self-harm, qualitative evidence synthesis, functions

## Abstract

**Background:** Eating disorders (EDs) and self-harm (SH) are both associated with distress, poor psychosocial functioning, and increased risk of mortality. Much of the literature discusses the complex interplay between SH and ED behaviours where co-occurrence is common. The onset of both is typically seen during teenage years into early adulthood. A better understanding of the functions of these behaviours is needed to guide effective prevention and treatment, particularly during the crucial developmental years. An earlier review has explored the functions of self-harm, but an equivalent review for eating disorder behaviours does not appear to have been completed. **Objectives:** This evidence synthesis had two objectives. First, to identify and synthesise published first-hand accounts of the reasons why people engage in eating disorder behaviours with the view to develop a broad theoretical framework of functions. Second, to draw comparisons between the functions of eating disorder behaviours and self-harm. **Methods:** A qualitative evidence synthesis reporting first-hand accounts of the reasons for engaging in eating disorder behaviours. A ‘best fit’ framework synthesis, using the a priori framework from the review of self-harm functions, was undertaken with thematic analysis to categorise responses. **Results:** Following a systematic search and rigorous screening process, 144 studies were included in the final review. The most commonly reported functions of eating disorder behaviours were distress management (affect regulation) and interpersonal influence. This review identified significant overlap in functions between self-harm and eating disorder behaviours. Gender identity, responding to food insecurity, to delay growing up and responding to weight, shape, and body ideals were identified as functions more salient to eating disorder behaviours. Similarly, some self-harm functions were not identified in the eating disorder literature. These were experimenting, averting suicide, personal language, and exploring/maintaining boundaries. **Conclusions:** This evidence synthesis identified a prominent overlap between psychological functions of eating disorder behaviours and self-harm, specifically in relation to distress management (affect regulation). Despite clear overlap in certain areas, some functions were found to be distinct to each behaviour. The implications for delivering and adapting targeted interventions are discussed.

## 1. Introduction

Eating disorders (EDs) have been described as “disabling, deadly, and costly mental disorders that considerably impair physical health and disrupt psychosocial functioning” [1] (p. 1). The primary features of EDs are noted to be disturbances in eating behaviours and accompanying cognitions. These behaviours include limiting food and fluid intake (restricting), eating excessive amounts of food in short periods of time (binge eating), overexercising, and compensating for eating through self-induced vomiting or laxative/diuretic use (purging) [2]. Self-harm (SH) refers to any intentional act of self-poisoning or self-injury, but clinical definitions often exclude behaviours that may lead to harm such as substance misuse or disordered eating [3]. Both EDs and SH are associated with distress, poor psychosocial functioning, and mortality [1,4,5,6].

There is a complex interplay between ED behaviours and SH where they frequently co-occur [7,8]. Commonalities include shared risk factors [7,9] and the age at onset, which are frequently seen in younger, often female, populations, notably teenage years into early adulthood [10,11]. Considering the overlap between the two, both researchers and clinicians have explored the potential for crossover in the reasons why people engage in these behaviours and to what extent ED behaviours can actually function as a form of SH [12]. Initial research suggested ED behaviours were an indirect form of SH where bodily harm was not the primary intention underpinning the behaviour [13,14]. However, subsequent work has shown that some people do describe engaging in disordered eating for the purpose of causing pain or damage to their body [15,16,17]. Clinically, there is growing recognition of how restriction can function as a form of self-harm. This has been termed RISH (Restrictive intake self-harm)—“a formulation-driven term that aims to describe the subset of patients who present with restricted intake (both foods and fluids) as a method of indirect self-harm [18] (p. 1).” This crossover between EDs and SH has, however, created challenges for those delivering clinical care, where misdiagnosis and uncertainty around treatment approaches are common [18,19]. Therefore, understanding more about the interaction between these behaviours may support clinical intervention and decision making. Current management guidelines have stressed the importance of psychological formulation approaches that takes account of the underlying functions of behaviour [19].

The specific functions underlying ED behaviours have not been explored to the same extent as those associated with SH. In the field of SH, conceptual frameworks such as Nock’s Four Function Model [20] and multiple reviews have provided a structured understanding of its functions [21,22]. The most recent is Edmondson’s 2016 [23] systematic review focused on self-reported accounts of non-suicidal reasons for SH, which presented 15 distinct functions underpinning SH. In contrast, whilst functional models exist for certain ED diagnoses [24,25,26,27], a comprehensive review of the functions of ED behaviours equivalent to those of SH appears to be lacking.

Existing research on the psychological mechanisms underpinning EDs primarily focuses on emotional regulation and weight loss, often framed within sociocultural ideals around striving for thinness [28]. However, a growing body of evidence suggests that ED behaviours may serve a range of other functions, such as communicating distress, creating a sense of control, and providing feelings of achievement or self-worth [27,29,30,31,32]. Despite this, there is currently no established framework designed specifically to assess the functions that motivate ED behaviours. Quantitative studies exploring potential transdiagnostic functions across SH and EDs have used measures originally developed for SH [12]. As a result, when such measures are applied to ED behaviours, important functional nuances specific to EDs may be overlooked.

There appears to be a gap in the literature for a specific framework of the functions of ED behaviours. Therefore, this evidence synthesis aimed to identify and summarise published first-hand accounts of the reasons why people engage in eating disorder behaviours (including restricting, binge eating, purging, and overexercising) with the view to develop a broad theoretical framework of functions. The second aim was to draw comparisons between the functions of eating disorder behaviours and self-harm.

## 2. Methods

Systematic review and qualitative synthesis of first-hand accounts of psychological functions of ED behaviours. This review was registered with PROSPERO, protocol registration number CRD42024555043. The methods within this review were guided by PRISMA guidelines [33]. For qualitative elements, the ENTREQ guidelines [34] were utilised.

### 2.1. Search Strategy

An exhaustive search was conducted on 11 July 2024 in the MEDLINE, Cochrane Library, CINAHL, PsycINFO, Embase, Web of Science, and ProQuest databases, supplemented with a Google Scholar search. Search strategy development was supported and peer reviewed by an information specialist based at the University of Leeds (GMJ) and utilised keywords categorised into three main areas relating to (i) ED behaviours, (ii) functions, and (iii) qualitative research. Search terms for the category of functions were developed based upon the previous review of non-suicidal reasons for SH [23]. Reference lists of all included studies were scanned to identify any further relevant papers. A detailed search strategy for Embase is available in Appendix A. Search algorithms for the other databases are available upon request.

### 2.2. Selection Criteria

The criteria for study inclusion can be seen below and are presented according to the SPIDER framework [35]:

*Sample*—Participants with self-reported ED behaviours as well as those with a current or historical diagnosis of an ED. For the purpose of this review, ED behaviours were limited to the following: restricting, binge eating, purging, and overexercising.

*Phenomenon of interest*—Functions of ED behaviours.

*Design*—Any qualitative method that elicited a self-reported account, attributed to an individual respondent and illustrated by direct quotations.

*Evaluation*—A clear specific self-reported account of reasons for engaging in ED behaviours.

*Research type*—Any primary, peer-reviewed, qualitative study available in the English language. Mixed method research studies were included only if they had a significant qualitative component.

#### Selection Criteria Justification

Both self-reported ED behaviours and clinically diagnosed EDs were included to try and encompass a wide range of perspectives, particularly as issues with defining and measuring EDs are well documented [36,37]. No other limits were placed on population, where ED behaviours have been shown to affect people of all ages, ethnicities, body weights, and gender identities [36,38,39]. The decision was made to include only recognised ED behaviours according to the ICD10 (International Statistical Classification of Diseases and Related Health Problems 10th Revision) criteria [40]. As a result, behaviours such as orthorexia and emotional eating, which are not recognised within this classification system, were excluded.

### 2.3. Paper Selection and Data Extraction

Citations derived from the search were exported into the reference management software Endnote X9 [41], at which point duplicates were removed and titles screened for any obvious exclusions, such as those not relating to EDs. Remaining citations were exported into Rayyan [42] where they underwent abstract, then full-text review to obtain the final list of included studies. All citations were reviewed by the primary author (F.A.), with second reviewers independently screening 50% of citations at the abstract phase (B.G. and S.M.) and 50% at full text (B.G.). Any discrepancies were resolved at a consensus meeting. Where discrepancies could not be resolved, other members of the review team were consulted (C.B., A.J.H., and T.A.W.).

Data on population, location, age, gender, research approach, aim of the study, and type of ED or ED behaviour were extracted from each included study using a standardised data extraction form. Reasons for ED behaviours were taken from direct quotations given by the authors of included studies. Generally, quotations provided an individual reason; however, occasionally more than one reason was present in a single quotation.

### 2.4. Data Synthesis

A ‘best fit’ framework synthesis utilising thematic analysis was undertaken to identify reasons for engaging in ED behaviours [43,44]. Guidance from the Cochrane Collaboration Qualitative Evidence Synthesis Handbook was utilised [45]. The a priori framework was based upon Edmondson’s review [23] of non-suicidal reasons for SH. The initial SH framework can be seen in Table 1.

The final dataset consisted of direct quotations extracted from studies meeting the eligibility criteria. Extracted quotations were initially allocated into the a priori framework, through repeated comparison of the dataset with the framework. Quotations added to each cell were constantly revisited as new data was added. Subsequent inductive thematic analysis was undertaken for quotations that did not fit the a priori framework. Quotations were initially clustered into themes, which were then iteratively discussed amongst the wider team until a consensus was achieved.

The primary author (F.A.) familiarised herself with the dataset and generated initial coding before meeting with other authors to identify areas of discrepancy. Generally, disagreements emerged around quotations that did not appear to fit the existing framework. Discussions were had regarding the development of new categories or refining existing categories. Discrepancies were resolved through discussion to reach consensus. Data management for the analysis was supported by the use of Microsoft Excel.

Results are expressed in terms of how many studies endorsed each function rather than how many individual quotations were present.

All authors were based at the University of Leeds during the conduct of this review and have diverse backgrounds and specialities. The main author (F.A.) is a white female with a background in mental health nursing and psychological therapy. Considering the potential of experiences and backgrounds to influence reviews of this nature, exploration of quotations and development of the final framework was iterative, consisting of wider discussions amongst the team to incorporate varying ideas and perspectives into the overall findings.

### 2.5. Quality Assessment

Quality assessment was carried out (independently by F.A.) on all studies that met the inclusion criteria, using the Critical Appraisal Skills Programme qualitative 10-point checklist [46]. Studies were categorised as low (score of 0–4/10), medium (score of 5–7/10) or high (score of 8–10/10). The aim of the quality assessment was to give an overall opinion on the strength of the evidence available but not to exclude studies based on quality. Much of the literature discusses the role of quality assessment in qualitive reviews where studies with identified methodological weakness still have the potential to provide rich and insightful data [47]. Where the aims of this review were to understand the nature and diversity of functions of eating disorder behaviours, it was felt that excluding studies based on quality could potentially lead to missed insights, where relevant data could potentially be retrieved from studies considered less ‘rigorous.’

### 2.6. Ethical Considerations

Whilst no new empirical data were gathered for the purpose of this review, the team were still mindful of potential ethical considerations and maintaining respect for the data provided by original participants. The review team are experienced in managing sensitive data and the dataset was not shared outside of the review team. With regards to informed consent, quotations were extracted from published studies in which participants had consented for their quotations to be used publicly. Only anonymised data were available via included studies and therefore the risk of individual identification of participants at review stage was minimal. Quality assessment processes also took account of ethical considerations within each individual study.

The sensitive nature of this topic and the potential for distress were frequently considered during the review process. The risk of researcher distress was mitigated through supervision and reflective discussions. Sensitive presentation of this research was a key consideration for the team, particularly around the use of language.

## 3. Results

The search yielded a total of 6345 citations. Following de-duplication, title, abstract, and full-text screening in addition to citation searching, a total of 144 eligible studies were included (Figure 1). The final review provided the results of qualitative analysis of 2938 participants with experience of ED behaviours. Some studies could not define participant numbers due to online anonymous data and therefore the total figure is likely to be larger than reported here.

Participant ages ranged from 10 to 76 years. For the purposes of this review, the following definitions were applied: child = 0–17, adult 18–65, and elderly 66 years+. Most studies were carried out in adult populations (100/144, 69%), followed by mixed child/adult (21/144, 15%), child (5/144, 3%), and mixed adult/elderly (3/144, 2%). One study investigated a mixed population that ranged from child to elderly and 14 studies did not report on ages of participants. Studies were carried out across 21 different countries; the majority explored samples from the UK, USA, and Australia (107/144, 74%). Sample sizes varied significantly from individual case studies to 360 online users (whereby Instagram posts were collected and analysed using inductive thematic analysis).

Whilst studies were carried out with participants identifying as a diverse range of genders (male, female, transgender, and non-binary), over half of all included studies were conducted with a female-only sample (84/144, 58%). Most studies utilised interviews as the primary method of data collection (116/144, 81%). Other methods included focus groups, open-ended questionnaires, social media posts, and participant written letters. A full table of all included studies and their characteristics can be found in Appendix A.

### 3.1. Quality Assessment Results

Most studies (134/144) were categorised as high quality, with 10/144 categorised as medium and 0/144 as low. This indicates that, overall, the studies in this review were generally deemed to be of high quality. As previously stated, no studies were excluded based on quality.

### 3.2. Final Framework of Reasons for ED Behaviours

Table 2 presents a final descriptive framework of self-reported reasons for engaging in ED behaviours. Differences between the original SH framework and the final ED framework were the following:Inclusion of five new functions (to die, responding to weight, shape and body ideals, responding to food insecurity, gender identity, and to delay growing up).Exclusion of four functions not endorsed by the ED literature (experimenting, averting suicide, personal language, and exploring/maintaining boundaries).Within the function of affect regulation, replacing the subcategory ‘managing physical over emotional pain’ with ‘responding to physical sensations.’Addition of the subcategory ‘managing loneliness and boredom’.Addition of ‘positive reinforcement from others’ to the function interpersonal influence.‘Taking up less space and/or disappearing’ and ‘avoiding demands of life’ as additional subcategories to the function of protection.Addition of subcategory ‘adhering to social norms’ to the function of belonging.Addition of subcategory ‘achievement/being good at something’ to the function of personal mastery.

A conceptual figure showing shared vs. distinctive functions of EDs and SH can be seen in Figure 2.

### 3.3. Functions Endorsed in the ED Literature That Were Also Present in the Original SH Framework

#### 3.3.1. Managing Distress (Affect Regulation)

Managing distress (affect regulation) was the most frequently endorsed function of ED behaviours, with over half of all included studies referring to this reason (82/144, 57%). This was consistent with the SH literature where this function was also the most widely researched. Frequently, ED behaviours were referred to as a coping mechanism:

“Over the years it [eating] developed into a distress management tool … [48]”, “It is a coping mechanism, and it is a ‘good one’ [49]”.

Within the original SH framework (Table 1), a common subcategory of dealing with physical rather than emotional pain was presented. However, this physical element was referred to in just a single ED study:

“I wanted the physical pain as opposed to the emotional because that made sense [50].”

In the original framework, SH was noted to function as a way of ‘cleansing the body’ or ‘letting out badness’ and this was also referenced in the ED literature:

“Anorexia offered this really clean, pure, serene, space [50]”, “The ritual of purging was like throwing out some evil or bad part of myself [51].”

Distraction was a common reason presented for SH in the original framework. This was also found in the ED literature:

“When I was concentrating about eating, I wasn’t thinking about anything else [52]”, “I think it occupies so much of your mind as well that you can’t really think about all the shit stuff [53].”

Frequently cited reasons within this function, which were not present in the original SH framework, were ‘responding to physical sensations’ and ‘managing loneliness and boredom.’

“Emotions for me, they express themselves in a physical way. I feel it in my stomach. […] I am scared of what will come, so I use food to suppress it [54]”, “The initial part of it was about being bored I would say [48].”

ED behaviours as a response to physical sensations were often described as a reaction to the body feeling full and an attempt to “empty myself” [55]. This reason appeared particularly salient for people experiencing ED behaviours who also had a diagnosis of autism.

“That feeling of putting on weight… that’s what kind of sends me back into restricting food, because it’s not about ‘oh god my stomach looks really big’, it’s more about I don’t like the sensation of how my stomach feels [56].”

#### 3.3.2. Exerting Interpersonal Influence

Forty-seven (33%) studies endorsed the function of interpersonal influence. Similar to the initial SH framework were subcategories of help seeking, communicating pain, and receiving love and affection.

“It’s my way of talking to the world. It tells everyone what I can’t. It allows me to show them how much I am hurting, how scared I am, how much I feel I am without [57]”, “You’re so tiny and vulnerable, and you get extra care [58]”, “The woman who was the skinniest got the most attention from my dad [51].”

In relation to help-seeking, several studies indicated how ED behaviours were often exacerbated by the idea that one was not unwell enough to deserve help:

“Oh well I’m not actually ill enough, I don’t actually deserve the help, I need to go out and be more anorexic [59].”

What appeared important to participants, that was not salient in the SH literature, was the way in which they were perceived by others, particularly with reference to romantic relationships and being desirable to others. ED behaviours therefore functioned as a way to gain the approval of others. This was a particularly significant function for female participants.

“It’s just fun to be skinny again, and all this attention from all these guys [60]”, “There was a positive reinforcement … indirectly … for not eating … starving myself … they would be like, yeah, keep continue what you are doing [61]”, “I was focused on being thin, pretty, and perfect in order to attract the prince who would provide me with love [62].”

#### 3.3.3. Punishment

Twenty-two (15%) studies supported the idea that ED behaviours were used as a form of punishment. Primarily this was presented as a way to punish the self:

“I didn’t eat. It was the way I could punish myself [63].”

Only two studies identified ED behaviours as a way to punish others:

“They would make me feel worthless, so I want to punish them by punishing myself, because I knew they would feel guilty over me punishing myself [30].”

#### 3.3.4. Dissociation

ED behaviours as a way to either bring on or terminate a dissociative state was reflected in 26/144 (18%) studies. Bringing on dissociation was more frequently endorsed than terminating a dissociative state. Often, participants referred to this as “self-medicating using food [64]” or “it’s like drugs, you manage not to think [58].”

“[F]or the most part, it stops me feeling. It numbs out pain, fear, anger, rejection [65]”, “If I’m restricting, my emotions are numb, so I don’t have to feel those emotions, it just numbs me from everything [66]”, “I felt more alert and present afterward [54].”

#### 3.3.5. Sensation Seeking

ED behaviours as generating a feeling of excitement or exhilaration were endorsed by 14 (10%) studies. Many quotations referred to getting a ‘high’ or a ‘buzz.’

“Succeeding [in carrying out a binge] was definitely exciting [67]”, “I used to get some perverse buzz out of not eating [68].”

#### 3.3.6. Expressing and Coping with Sexuality

Consistent with the SH framework, ED behaviours serving a sexual function was a low endorsed reason, with only six (4%) studies referring to this. Whilst the SH literature found evidence to suggest SH as providing something similar to a ‘sexual release’ this was not apparent within the ED studies. Instead, quotations referred to either becoming more or less interested in sex. Some referred to ED behaviours as being an important element to their sexual experiences:

“It was the skinnier I got the more I could flaunt myself, the more sexually promiscuous I became [69]”, “It stops me caring about men and wanting relationships or sex or whatever [65], “I love being made to feel small and delicate by my sexual partners, especially masculine ones [70].”

Meanwhile, others referred to ED behaviours as a way to deny their sexuality by avoiding subjectivities associated with their sexual orientation:

“I was trying to keep myself, looking (.) straight maybe [71],” “I just saw being thin as being straight [71].”

#### 3.3.7. Gratification

Thirty-six (25%) studies reported the positive function of ED behaviours as providing a sense of happiness or comfort:

“Food is the only thing that makes me feel good. It’s always there and it comforts me [72],” “I find [binge eating and heavy drinking] gives me comfort and warmth. It makes me feel fulfilled [73].”

#### 3.3.8. Protection

ED behaviours serving the function of protection was apparent in 43/144 (30%) studies. The majority of these focused on protecting the self with only one study referencing the protection of others. This was in relation to protection from the individual’s poor emotional state that occurred when they did not engage in ED behaviours.

“Since I can be really short-tempered, I feel I will be awful to be with if I do not exercise [74].”

Within the function of protecting the self, avoiding unwanted attention/sexual advances was a salient subcategory and was often made in reference to previous trauma.

“Being overweight helped me later to stop him from touching me [75]”, ‘‘I was an attractive child, and I felt uncomfortable with adoration; always been told that I was pretty. So being told that I was disgustingly thin was better than being beautiful [76].”

Safety appeared to be an important element within this function, with many quotations referring to ED behaviours as a ‘friend’ or a ‘safe place’:

“I think it [anorexia] becomes a bit of like a friend, like it’s a, it’s almost like it’s a world that you live in that’s separate from everybody else [53],” “It was another world, you know, it was like just a safe place [77].”

Other important elements, which did not appear prominent within the SH review, were protecting oneself by taking up less space and ED behaviours offering people a way to avoid the demands of life or being something to blame when things went wrong. It was considered whether this subcategory would be better placed within interpersonal influence, as often quotations referred to avoiding social interactions. However, it was felt that avoiding obligations ultimately served the purpose to protect the person from things they felt they could not cope with.

“I wanted to ‘disappear’ become smaller and less noticeable [70]”, “Anorexia can be a good excuse to keep me away from others. It can act as a get out clause, an excuse [59],” “It was like my own way of withdrawing, protecting myself from people getting close to me, [and of] not having obligations [77].”

#### 3.3.9. Validation

ED behaviours as a form of validation were endorsed by 56 studies (39%). In line with the SH literature, ED behaviours were described as a way to demonstrate strength or toughness.

“It proved to myself that I wasn’t, that I wasn’t weak, you know, that I could make myself do that [vomit] … [78]”, “Having the lowest calorie meal on the menu made me feel really strong [79].”

Similar to the original SH framework was a sense of self-validation in comparison to others, with participants expressing that their ED behaviours made them special, unique, and different.

“You make me feel special by making me different. You give me something that none of my friends or family have [80].”

There was an element of competition within this, where participants expressed the desire to be “the best anorexic [81,82]”:

“Are they thinner than me? Am I thinner than them? And I found it very, very competitive [66].”

One reason that stood out within this category, which differed from the SH literature, was ED behaviours as an achievement. Thirty-eight studies referred to ED behaviours providing a sense of achievement or ‘being good at something’:

“It gives you something to aim for, it’s the only thing I’ve ever done that makes me feel like I’m good at something [52],” “The only goal I seemed to have was to not eat and be thin and so when I looked in the mirror I was […] triumphant […] because I’d think oh yeah look at all you’ve achieved [83].”

#### 3.3.10. Belonging

A total of 35/144 (24%) studies referred to ED behaviours being used to achieve a sense of belonging or acceptance:

“I could throw up my food, lose some weight and maybe if I was thin then I’d feel like I belonged [84]”, “When I am thin, I will not be judged by my appearance. I will be accepted [85]”.

Many studies referred to eating disorders and ED behaviours as an identity:

“I have never had much of a sense of self, and I think possibly [anorexia] then became a little bit like an identity [56].”

Multiple quotations referred to ED behaviours as a way of adhering to cultural or social norms. We placed this within the category of belonging, but recognise how this relates to broader societal standards and expectations, which may differ across cultures and contexts:

“I feel like women have to be quite self-conscious, erm, in my culture anyway, because—well, you know, needing to be attractive enough to be married off to a man [86].”

#### 3.3.11. Personal Mastery

ED behaviours as providing a sense of personal mastery was the second most endorsed function within the review (77/144, 53%). The majority of these papers spoke about ED behaviours providing a sense of control whilst others referenced certainty, routine, and structure.

“I just felt I needed control and that (restrictive eating) was one way of controlling at least part of my life [87]”, “you [anorexia] put the messiness of life into tiny compartments, each one boxed and labelled and managed in turn [88].”

### 3.4. Emergent Functions More Salient in the ED Literature

Several studies included within this review identified one or more reasons that did not fit into the original framework based on SH. These include the following.

#### 3.4.1. To Die

ED behaviours with an eventual goal of ending one’s life were endorsed by five studies: “My biggest goal is to starve myself to death, that’s what I’m longing for [58]”. Whilst the reason of dying was noted as a prevalent reason for SH, it was not part of the original SH framework, where Edmondson’s review [23] aimed to explore nuanced understandings of motives for SH outside of discussions focusing on suicidal intent. Therefore, whilst reported as a new function within this review, ED serving as a function ‘to die’ should be noted as overlapping with SH. Only a very small number of ED studies endorsed this function; this therefore appears to be a more salient function for SH rather than ED behaviours.

#### 3.4.2. Responding to Weight, Shape, and Body Ideals

ED behaviours that function to manage concerns about one’s body, weight, and shape were frequently cited as reasons amongst the included studies, and these were a function distinct from the SH framework.

“A big part had to do with weight and shape issues, to help me lose weight [89].”

Within this function a commonly cited reason was to find balance (using ED behaviours to eat without feeling guilt or putting on weight).

“I thought, well this is so great, I could eat all I want and not gain any weight [51]”, “So, instead of restricting for so long it was like, ‘great I can eat this at dinner time and just throw it back up’ [30]”.

#### 3.4.3. Responding to Food Insecurity

Six (4%) studies referred to managing food insecurity as a reason for ED behaviours, both in relation to restricting and binge eating:

“When there’s not enough money it’s easier to skip out … way easier to restrict or under portion [90]”, “I would overeat, we always overeat … there were times when it felt like there was no food in the house [91].”

#### 3.4.4. Gender Identity

Ten (7%) studies discussed functions in relation to managing gender dysphoria and adhering to or going against gender societal expectations.

“Size is gendered … the thinner you are, the more feminine … [and] I wanted to be read more as female [92]”, “I guess the answer is I relate to my gender by starving myself because I feel more gender affirmed [90]”, “I didn’t feel at ease with my body, maybe because what I was seeing was still a male body and what I wanted was a female one, a more slender and less muscular one. I decided to keep losing weight so that I could feel more like myself [93].”

#### 3.4.5. To Delay Growing up

Whilst only a small number of studies (6/144, 4%) referred to a reason being to delay growing up, it was felt to a be a distinct and important category to report:

“In the face of the possibility of puberty, I had this fear that if I started showing signs of ‘growing up’ it might make my parents feel ‘old.’ I did not want my parents to feel old or like they were losing me, so I sought to ‘fight puberty’ by not eating [57].”

Discussions were had amongst the team as to whether such quotations were more aligned with functions of interpersonal influence or gender identity, particularly when referring to wanting to remain a child, to receive love/attention, or to delay puberty due to gender dysphoria. However, it was felt this function should be reported as a standalone category as it appears to be distinct from the SH literature.

### 3.5. Functions More Salient in the SH Literature

#### 3.5.1. Averting Suicide

No studies were found in this review that supported the idea of ED behaviours serving the function of averting suicide. Whilst four studies did refer to ED behaviours as a way to survive, it was felt that this referred more to ED as a coping mechanism rather than to directly stop someone acting on suicidal thoughts.

#### 3.5.2. Maintaining or Exploring Boundaries

Only one study endorsed ED behaviours functioning to maintain or explore boundaries. With only one study referring to this function, it was felt that this did not appear to be a salient function for ED behaviours.

“It defined the blurry edges of my being with clear, hard straight lines; no diffuse, wandering self, no doubts about what I was, where I started and ended, I became fixed and unfluctuating [55].”

#### 3.5.3. Experimenting

The function of experimenting or ‘trying something new’ was not endorsed by any of the studies included within this review.

#### 3.5.4. Personal Language

ED behaviours as a private language, used as a means or conjuring up or acknowledging past feelings or memories did not appear in any of the included studies.

## 4. Discussion

This review aimed to explore and categorise the functions of ED behaviours through the lens of an existing SH framework. The findings highlight both the substantial overlap as well as clear distinctions between the two spectra of behaviours, demonstrating the need for an ED-specific framework. Many reasons for ED behaviours fit within the a priori SH framework (Table 1), offering support to the existing body of literature that documents the overlap and transdiagnostic functions between SH and ED behaviours [8,12,17,94,95]. The findings of this review suggest that both behaviours are motivated by similar functions, which could help to explain why they are frequently seen as co-occurring and interchangeable coping strategies [94].

Consistent with the SH literature, the most widely researched function in relation to ED behaviour was managing distress (affect regulation), highlighting this as a core function for ED behaviours [32,96,97]. This supports the literature, which identifies emotional regulation as a transdiagnostic process across various psychopathologies [98,99]. Unlike the SH framework, the ED literature emphasised responses to physical sensations and sensory distress pointing towards a sensory regulation function, particularly for neurodivergent individuals [56]. Another frequently endorsed function that aligned with SH was interpersonal influence. The crossover was particularly relevant for areas of help seeking, communicating distress, and receiving love/attention from others. However, differences within this function emerged, notably the utilisation of ED behaviours to elicit approval or positive reinforcement from others, often in romantic contexts. This difference may be reflective of wider sociocultural ideals around thinness, attractiveness, and femininity, which are not in the same way present for SH [100,101].

There were key differences between the most endorsed functions for SH relative to ED behaviours. Punishment and dissociation were two of the most frequently cited reasons for SH, but this was not the case for ED behaviours, where reasons such as validation, personal mastery, protection, and belonging were more frequently referenced. These differences may be unsurprising when considered in the context of research that describes SH as a more private/intrapersonal experience than ED behaviours [102], which are more likely to be influenced by social comparison and a desire to conform to societal norms [103,104]. Such ideas may begin to explain why functions around control, validation, and belonging appear to be much more salient in the ED literature when compared with SH.

Four a priori SH functions did not arise in any of the included ED studies. Similarly, quotations were found in the ED literature that did not correspond to any SH function. This indicates that functions exist that are more salient to each. Notably, the function of averting suicide was not supported by any quotations from the ED literature, and this aligns with other research that has been conducted in the area [12]. One distinct and related theme did emerge, that of using ED behaviours as a means to end one’s life. This potentially speaks to research that identifies increased risk of suicide in those experiencing EDs [105,106] and challenges conceptualisations of ED behaviours as non-suicidal. Whilst it was not one of the more salient functions within the ED literature, indicating it is more closely related to SH, it was nevertheless an endorsed reason for why some people engaged in ED behaviours.

Other distinctive ED functions identified within this review included managing weight, shape and body ideals, gender identity, to delay growing up, and managing food insecurity. The first of these is perhaps the most widely recognised function of ED behaviours (considered core psychopathology) and a key difference from motivations around SH. The other novel functions point towards the need for gendered and sociocultural considerations when understanding the functions of ED behaviours, where several papers referred to using ED behaviours as a way of “performing femininity or masculinity” [92,107,108]. This potentially speaks to gender schema theory [109], which suggests that children develop knowledge (schemas) about what it means to be a ‘male’ or ‘female’ through observing the world around them. These schemas shape understanding of gender norms/expectations and thus influence how people behave. This function appeared particularly relevant for transgender and gender diverse individuals who frequently discuss using ED behaviours as an attempt to suppress or alter their physical characteristics in an attempt to feel more gender-aligned [70,90,92,93,108,110,111]. This overlapped with the function of ‘to delay growing up,’ suggesting that attention should be given to identity formation during key developmental phases in children and young adults. Lastly, food insecurity was a distinct finding from the SH literature and reinforces recent research that has started to explore the relationship between food insecurity and disordered eating behaviours [112,113,114].

During the iterative phases of data synthesis, the functions of gender identity, sexuality, to delay growing up, managing distress, and interpersonal influence were noted to be closely inter-connected. An example is to delay growing up, linked closely with interpersonal influence, whereby participants felt that they might receive more love and attention if they were seen as a ‘child.’ Gender identity further linked to delay growing up, whereby some people described using ED behaviours to disrupt puberty and ultimately delay their body physically developing in a way that was associated with certain genders. Gender identity and sexuality linked to interpersonal influence as well as managing distress, where ED behaviours were utilised to mask sexual orientation or deal with negative comments/stereotypes associated with gender identity and/or sexual orientation. This demonstrates the importance of unpicking functions of ED behaviours specific to the individual, which could help to guide psychological interventions accordingly.

While many functions were endorsed across a range of ED behaviours, some appeared to be more closely associated with specific behavioural types. For instance, binge eating and purging were frequently linked with loneliness and boredom, suggesting these behaviours play a role in providing stimulation or ‘filling emotional voids/gaps.’ In contrast, quotations referencing positive reinforcement and validation from others predominantly related to restricting, highlighting how thinness and notions of self-discipline are socially praised and ultimately reinforced [103,104]. Functions of gratification and sensation seeking were almost exclusively associated with binge eating. The function of control was most commonly associated with restriction, though some quotations referred to purging and overexercise. Notably, binge eating was absent from this category, which may reflect the literature describing loss of control as a defining feature of bulimia nervosa and binge eating disorder, but not necessarily of other ED diagnoses, which are primarily characterised by restrictive behaviour [115,116]. Such differences across the varying behaviours highlight the diversity of ED behaviours and the complex interplay of functions.

### 4.1. Strengths and Limitations

This was a comprehensive and rigorous review analysing a large number of qualitative studies with functions found across many contexts. While the analytical approach aimed to remain grounded in participants’ lived experiences, it required a level of judgment removed from the original context of the transcripts. Therefore, there is the risk that the meaning of some participant quotations may have been altered or misunderstood when extracted from their broader context. Additionally, this review relied on the selection and presentation of data by original study authors, including their decisions about which quotations to include and how to frame them. This is a particular limitation of using secondary data and could have influenced the synthesis or resulted in a partial representation of perspectives. As such, these results should be viewed as interpretative, and future research could focus on validating and refining the proposed framework alongside people with lived experience.

Frequency counts and percentages were included as a proxy indicator of theme saturation across the dataset, which is consistent with Edmondson’s review [23]. However, caution should be taken when interpreting these percentages for the following reasons. The aims of included papers were heterogenous and therefore some functions, whilst potentially salient to participants, may not have been articulated due to the scope of the paper and questions asked. In addition, firsthand accounts were carefully selected by authors of the included articles and are likely to have been limited by factors such as word counts. Therefore, it is not possible to precisely quantify the saturation of themes across the literature as a whole. However, percentages do offer an indication of how often each function came up.

Whilst efforts were made to differentiate between distinct ED behaviours, this was not always possible. Many of the included studies/quotations referred to behaviours in relation to broader terms, e.g., anorexia, which has the potential to encompass a range of behaviours. Consequently, the commentary made on reasons in relation to types of behaviour can only be seen as partial, and it is possible that subtle associations have been missed. Additionally, included studies predominantly reflect Western perspectives and many focused on the perceptions of women. There is the potential for culture/region to influence functionalities of ED behaviours across different populations; however, comprehensively exploring this was something that was outside of the scope of this review. Therefore, future research that explores how cultural and contextual factors influence the functionality of ED behaviours may be beneficial.

### 4.2. Implications

This review has presented a theoretical framework outlining the functions of ED behaviours, offering a foundation for understanding the diverse and often personal reasons individuals engage in them. This framework could support clinicians in assessing ED behaviours, aiding in case formulation and the development of tailored management plans that address the individual’s unique motives and goals. EDs are notoriously difficult to treat and are associated with the highest mortality among all mental disorders [117]. The literature has noted difficulties in developing effective interventions for EDs [118,119] and this may be partly due to a limited understanding of functions of ED behaviours. ED behaviours are often seen as beneficial and protective by those who engage in them, frequently leading to ambivalence around recovery [57]. Therefore, therapeutic approaches that acknowledge the perceived beneficial and protective functions of ED behaviours and collaboratively support individuals in developing alternative strategies that fulfil similar roles may enhance engagement and treatment outcomes.

The commonly endorsed functions from this review offer a foundation for developing targeted interventions focusing on motivations underlying ED behaviours. For example, treatments centred around emotional regulation, interpersonal relationships, and identity formation may be particularly beneficial in addressing core functions. However, this review has also highlighted the deeply personal nature of ED behaviours and the wide range of functions they may serve. As such, treatment should be holistic and person-centred, with the potential for this framework to be utilised and adapted to help identify the functions most salient to each person.

Through using an a priori framework based upon self-harm, this review has also been able to compare the functions underlying ED behaviours and SH. The findings suggest that, whilst distinctions exist, there are many shared functions between the two behaviours. This has potential implications for the treatment of those who experience both SH and EDs, where the literature has indicated a lack of interventional studies that target these behaviours concurrently. A deeper understanding of the functions that are reflective of both behaviours offers a basis for function-based intervention development for people who engage in both ED behaviours and SH. Where clinicians identify overlapping functions, there is the potential for treatment to target shared functions, which may lead to the effective reduction in both behaviours.

## 5. Conclusions

This review has demonstrated the multiple and varied self-reported reasons why people engage in ED behaviours. Our findings indicate a prominent overlap between reasons for ED and SH behaviours, strengthening the empirical basis for an association between the two and indicating where interventions may target these functional similarities. Future research should focus on validating this framework alongside people with lived experience and supporting relevant intervention development.

## Figures and Tables

**Figure 1 healthcare-13-01914-f001:**
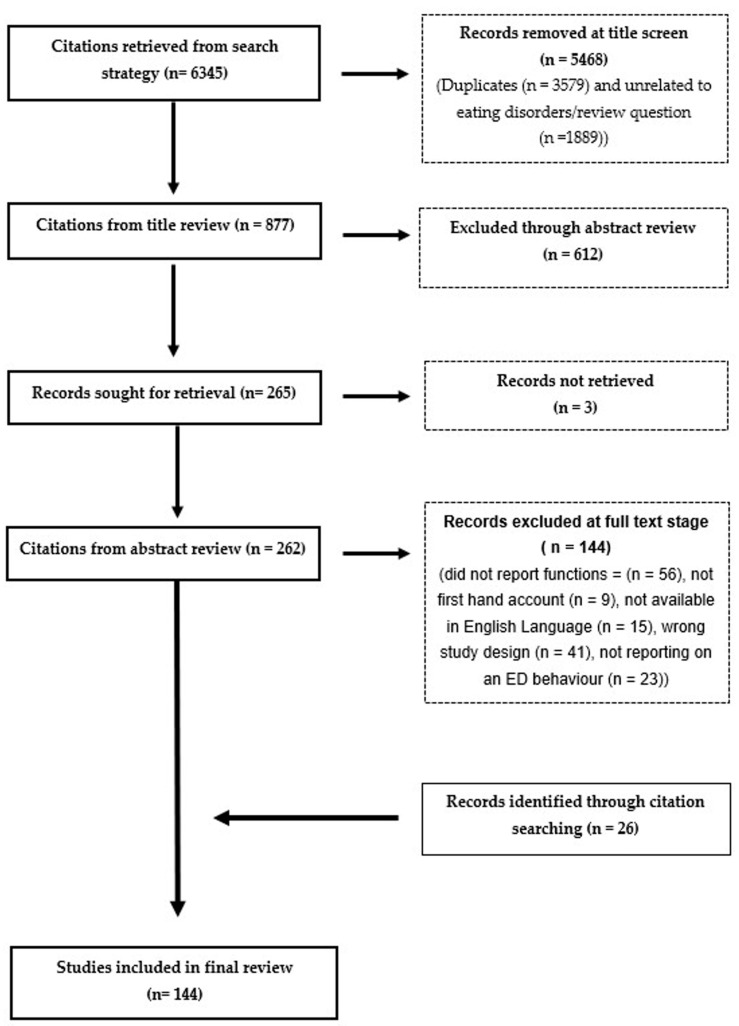
Flow chart of included and excluded studies.

**Figure 2 healthcare-13-01914-f002:**
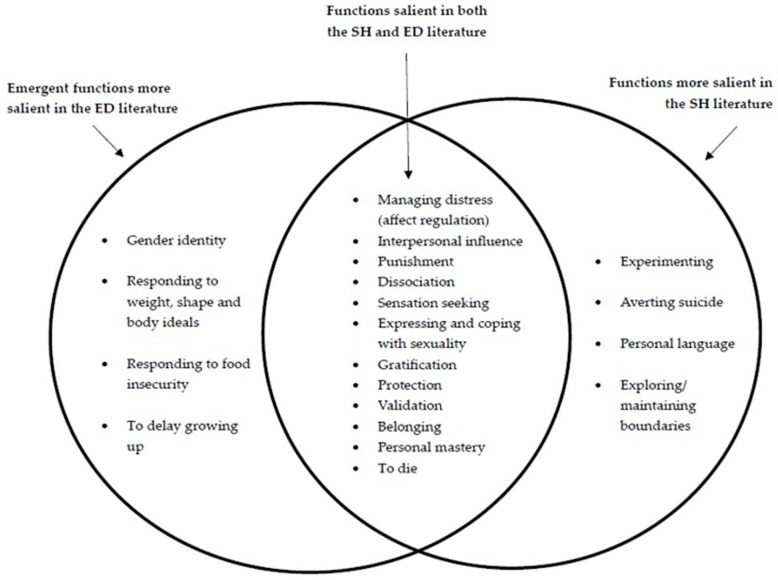
Conceptual figure of shared vs. distinctive functions of EDs and SH.

**Table 1 healthcare-13-01914-t001:** A priori framework—non-suicidal reasons for self-harm.

Responding to Distress	Self-Harm as a Positive Experience	Defining the Self
**Managing distress (affect regulation)**—managing painful unpleasant emotional states including making emotional pain physical; blocking bad memories	**Gratification**—self-harm as comforting or enjoyable	**Defining boundaries**—self-injury as a means of defining or exploring personal boundaries
**Interpersonal influence**—changing or responding to how others think or feel; help-seeking	**Sensation seeking**—through a sense of non-sexual excitement or arousal	**Responding to sexuality**—through self-harm as creating quasi-sexual feelings or expressing sexuality in a symbolic way
**Punishment**—usually of the self, occasional of or by others	**Experimenting**—trying something new	**Validation**—demonstrating to the self and occasionally to others one’s strength or the degree of one’s suffering
**Managing dissociation**—either switching off or bringing on feelings of numbness and unreality	**Protection**—of self or others	**Self as belonging or fitting in**—to a group or subculture
**Averting suicide**—nonfatal self-harm to ward off suicidal acts or thoughts	**Develop a sense of personal mastery**	**Having a personal language**—including one for remembrance: a means of conjuring up or acknowledging good past feelings or memories

**Table 2 healthcare-13-01914-t002:** Final theoretical framework of reasons for eating disorder behaviours.

Responding to Distress	Self-Harm as a Rewarding Experience	Defining the Self
**Managing distress** (affect regulation)—managing painful unpleasant emotional states including distraction, cleansing, release, dealing with physical sensations, and managing loneliness or boredom	**Protection**—primarily of the self but occasionally of others. Eating disorder behaviours as a friend or safe space that helps the person to avoid unwanted attention, avoid the demands of life, and take up ‘less space’	**Responding to weight, shape and body ideals**—responding to feelings of unhappiness in relation to the way one looks, particularly in relation to body norms and ideals. To find balance, make up for eating, or to eat without perceived negative consequences
**Interpersonal influence**—responding to how others think or feel, communicating pain, help seeking and positive reinforcement from others	**Sensation seeking**—through a sense of non-sexual excitement or arousal	**Responding to sexuality**–expressing sexuality in a symbolic way
**Punishment**—primarily towards the self but occasionally towards others	**Gratification**—eating disorder behaviours as comforting or enjoyable	**Validation**—demonstrating strength or degree of one’s suffering. Feelings of achievement or being good at something
**Managing dissociation**—generally to bring on feelings of numbness but sometimes to break a dissociative state	**Developing a sense of personal mastery**—to feel in control and gain a sense of certainty, routine, and structure	**Belonging or fitting in**—to a group or subculture and adhering to social norms
**To die**—eventual goal of ending one’s life		**Gender identity**—managing gender dysphoria or affirming gender identity
**Responding to food insecurity**—restricting or overeating in relation to food insecurity		**To delay growing up**—to avoid responsibilities of being an adult or access love/care that was received as a child

## Data Availability

Not applicable.

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
