# Peer review of "How Do the Psychological Functions of Eating Disorder Behaviours Compare with Self-Harm? A Systematic Qualitative Evidence Synthesis"

_healthcare, 2025, doi:10.3390/healthcare13151914_

Round 1

Reviewer 1 Report

Comments and Suggestions for Authors

Dear Authors,

Thank you for the opportunity to review your manuscript, “How Do the Psychological Functions of Eating Disorder Behaviours Compare with Self-Harm? A Systematic Qualitative Review.” Your work addresses an important clinical and theoretical issue and is generally well executed. The identification of overlapping and distinct functions of ED and SH behaviours is of considerable clinical relevance.

However, I recommend the following major and minor revisions, indicated below with specific line numbers, to further improve clarity, rigour, and alignment with reporting standards (e.g., PRISMA, ENTREQ).

COMMENT 1: Methodological transparency
Lines 91–165:

  • Clarify whether any qualitative analysis software (e.g., NVivo) was used.
  • Indicate how inter-coder reliability was managed.
  • Describe how the team resolved disagreement

COMMENT 2: Lines 153–165:

Explain how themes were refined iteratively (e.g., use of matrices or thematic clustering).  

Clarify the level of abstraction used to define the 11 higher-order themes.

COMMENT 3: Lines 465–547 (Discussion):

Consider integrating models such as transdiagnostic frameworks of emotion regulation, gender schema theory, or functional analysis to deepen discussion.

Consider adding a conceptual figure summarising shared vs. unique ED/SH functions.

COMMENT 4: Abstract format
Lines 9–37:

While the content is clear, MDPI typically encourages structured abstracts (Background, Objectives, Methods, Results, Conclusion). Consider reformatting accordingly.

COMMENT 4: Terminology consistency

Lines 22, 25, 33, 535:

Use consistent terminology when referring to ED behaviours (e.g., “restricting,” “bingeing,” “purging,” “overexercising”) throughout the manuscript and figures.

Correct minor phrasing issues (e.g., “to delaying growing up” → “to delay growing up”; ensure singular/plural agreement).

I encourage you to consider the points raised in this review as an opportunity to enhance both the scholarly value and clinical impact of your contribution. I look forward to seeing your revised submission.

Best Regards,

Author Response

Dear Reviewer 1,

Thank you very much for taking the time to review this manuscript. Please see below a point-by-point response to your comments in red text. Corresponding revisions/corrections can be seen in tracked changes in the re-submitted files. 

Reviewer Comments:

Dear Authors,

Thank you for the opportunity to review your manuscript, “How Do the Psychological Functions of Eating Disorder Behaviours Compare with Self-Harm? A Systematic Qualitative Review.” Your work addresses an important clinical and theoretical issue and is generally well executed. The identification of overlapping and distinct functions of ED and SH behaviours is of considerable clinical relevance.

However, I recommend the following major and minor revisions, indicated below with specific line numbers, to further improve clarity, rigour, and alignment with reporting standards (e.g., PRISMA, ENTREQ).

COMMENT 1: Methodological transparency
Lines 91–165:

  • Clarify whether any qualitative analysis software (e.g., NVivo) was used.
  • Indicate how inter-coder reliability was managed.
  • Describe how the team resolved disagreement

Reply: Thank you very much for your helpful feedback. Qualitative analysis software was not utilised in this review and was conducted manually using Microsoft Excel. The following statement has been added to the manuscript to ensure transparency:

“Data management for the analysis was supported by the use of Microsoft Excel.”

With regards to inter-coder reliability and resolution of team disagreements the following paragraph has been added to the manuscript:

“The primary author (FA) familiarised herself with the dataset and generated initial coding before meeting with other authors to identify areas of discrepancy. Generally, disagreements emerged around quotations that didn’t appear to fit the existing framework. Discussions were had regarding the development of new categories or refining existing categories. Discrepancies were resolved through discussion to reach consensus.”

COMMENT 2: Lines 153–165:

  • Explain how themes were refined iteratively (e.g., use of matrices or thematic clustering).  
  • Clarify the level of abstraction used to define the 11 higher-order themes.

Reply: The following paragraph has been added to the manuscript to provide clarity around the analysis and how themes were refined iteratively. 

“Extracted quotations were initially allocated into the a priori framework, through repeated comparison of the dataset with the framework. Quotations added to each cell were constantly revisited as new data was added. Subsequent inductive thematic analysis was undertaken for quotations that did not fit the a priori framework. Quotations were initially clustered into themes which were then iteratively discussed amongst the wider team until a consensus was achieved.”

Apologies, we were a little unsure about the second comment around level of abstraction used to define the 11 higher-order themes. We think you may be referring to the 11 functions/themes which were endorsed in the ED literature and were also present in the original SH framework. If so, these themes were part of the a priori framework extracted from Edmondson’s review of functions of SH. The team placed extracted quotations into these themes based upon where they felt they were best aligned. Those that did not fit were utilised to create new ED specific themes. We hope this helps clarify and apologies if we have misunderstood this comment.

COMMENT 3: Lines 465–547 (Discussion):

  • Consider integrating models such as transdiagnostic frameworks of emotion regulation, gender schema theory, or functional analysis to deepen discussion.
  • Consider adding a conceptual figure summarising shared vs. unique ED/SH functions.

Reply: Thank you very much for this feedback and the opportunity to deepen the discussion section of the paper. Literature around emotional regulation as a transdiagnostic process and gender schema have been added into the discussion (Lines 529-530 and 565-570). Functional analysis/models of ED behaviour have been noted also; however, this has been added to the introduction rather than discussion section (Line 75).

A conceptual figure summarising shared vs distinct ED/SH functions has been added.

COMMENT 4: Abstract format
Lines 9–37:

While the content is clear, MDPI typically encourages structured abstracts (Background, Objectives, Methods, Results, Conclusion). Consider reformatting accordingly.

Reply: Thank you for informing us regarding the formatting of the abstract. This manuscript has been amended accordingly to add in an objectives section.

COMMENT 5: Terminology consistency

Lines 22, 25, 33, 535:

Use consistent terminology when referring to ED behaviours (e.g., “restricting,” “bingeing,” “purging,” “overexercising”) throughout the manuscript and figures.

Correct minor phrasing issues (e.g., “to delaying growing up” → “to delay growing up”; ensure singular/plural agreement).

Reply: Many apologies for the inconsistencies in terminology.  The manuscript has been reviewed to ensure consistent terminology, and the following terms have now been used throughout:

  • Restricting
  • Binge eating
  • Purging
  • Overexercise

In line 535, “to delaying growing up” has been changed to “to delay growing up.”

Reviewer 2 Report

Comments and Suggestions for Authors

Thank you for the opportunity to review this interesting article.

The article compared psychological functions of those with eating disorders and those who committed self-harm through a systematic review.

The topic is important and relevant to clinical practice. The methodology is rigorous. The reporting of results is very thorough, especially the Table summarizing study characteristics. The findings are useful.

Below are my minor suggestions to further improve the paper:

1) Inclusion and exclusion criteria: Please define clearly what are inclusion and exclusion criteria, ideally using the PECO(s) framework or lists. For example, are those who committed self-harm included only or those who considered or attempted self-harm also included? These should be clarified.

2) Figure: Figure resolution can be improved with vector format, such as svg.

3) Percentage: I am a bit suspicious about counting percentages of papers that discuss the theme. There are two possibilities: first, there is no relationship found in the study (which seems to be your assumption), and second, the variable is beyond the scope of the study. I would acknowledge this limitation for readers to accurately interpret the percentages.

Best luck moving forward!

Author Response

Dear Reviewer 2,

Thank you very much for taking the time to review this manuscript. Please see below a point-by-point response to your comments in red text. Corresponding revisions/corrections can be seen in tracked changes in the re-submitted files.

Reviewer Comments:

Thank you for the opportunity to review this interesting article.

The article compared psychological functions of those with eating disorders and those who committed self-harm through a systematic review.

The topic is important and relevant to clinical practice. The methodology is rigorous. The reporting of results is very thorough, especially the Table summarizing study characteristics. The findings are useful.

Below are my minor suggestions to further improve the paper:

1) Inclusion and exclusion criteria: Please define clearly what are inclusion and exclusion criteria, ideally using the PECO(s) framework or lists. For example, are those who committed self-harm included only or those who considered or attempted self-harm also included? These should be clarified.

Reply: Thank you very much for your kind feedback. We agree that the use of a framework is a helpful addition which makes the criteria much clearer to understand. We did explore the PECO(s) framework initially. However, in the end we decided to utilise the SPIDER tool. We felt that the SPIDER headings were more aligned to our review question which didn’t include treatments/interventions (exposure) or comparators. The selection criteria section of the manuscript has been rewritten using SPIDER as a framework.

With regards to people who self-harm, this was the population of focus of Edmondson’s review that the a priori framework was based upon. Therefore, it is not specifically included in our selection criteria, which focused solely on ED behaviour. For clarity Edmondson’s review describes taking “a broad approach to the definition of self-harm as a behaviour - any intentional act of self-poisoning or self-injury, (NICE, 2004) excluding only indirectly self-harming behaviour such as harmful alcohol or drug use, and regardless of the method of self-harm used.” Their inclusion criteria therefore refers to self-harm as an act (irrespective of motive e.g. suicidal or not) rather than ideation only.

2) Figure: Figure resolution can be improved with vector format, such as svg.

Reply: Thank you for identifying this alternative format which could improve figure resolution. I have changed both figures in the paper to svg. format. Could I kindly ask the editors if this is an acceptable format for the journal?

3) Percentage: I am a bit suspicious about counting percentages of papers that discuss the theme. There are two possibilities: first, there is no relationship found in the study (which seems to be your assumption), and second, the variable is beyond the scope of the study. I would acknowledge this limitation for readers to accurately interpret the percentages.

Reply: Thank you for raising this. We absolutely agree with this point and the need for this to be better articulated as a limitation within the paper. Therefore, the following paragraph has been added to the manuscript to reflect this:

Frequency counts and percentages were included as a proxy indicator of theme saturation across the dataset which is consistent with Edmondson’s study. However, caution should be taken when interpreting these percentages for the following reasons. Firstly, the aims of included papers were heterogenous and therefore some functions, whilst potentially salient to participants, may not have been articulated due to the scope of the paper and questions asked. In addition, firsthand accounts were carefully selected by authors of the included articles and are likely to have been limited by factors such as word counts. Therefore, it is not possible to precisely quantify the saturation of themes across the literature as a whole. However, percentages do offer an indication of how often each function came up.

Reviewer 3 Report

Comments and Suggestions for Authors

Dear authors, thank you for the opportunity to review your work. I found it very interesting.

This is a well-conducted systematic review.

Here are some guidelines that I think could improve your work:

  • Include other systematic reviews that address eating disorders and/or self-harm behaviors.
  • PLEASE RECORD THE PROTOCOL FOR THIS PAPER. THIS PAPER CONCERNS, IN ADDITION TO EATING DISORDERS, THEIR RELATIONSHIP WITH SELF-HARMING BEHAVIORS. ​​The protocol indicated corresponds to previous work.
  • If possible, include the search algorithms used for the remaining databases in the Supplementary Material – Search Strategy. Verify that they work correctly.

Author Response

Dear Reviewer 3,

Thank you very much for taking the time to review this manuscript. Please see below a point-by-point response to your comments in red text. Corresponding revisions/corrections can be seen in tracked changes in the re-submitted files.

Reviewer comments:

Dear authors, thank you for the opportunity to review your work. I found it very interesting. This is a well-conducted systematic review.

Here are some guidelines that I think could improve your work:

  • Include other systematic reviews that address eating disorders and/or self-harm behaviours.

Reply: Thank you very much for your kind feedback. The following reviews (although not all systematic) relating to eating disorders and/or self-harm behaviours are referenced within the paper.

  • Carroll, R.; Metcalfe, C.; Gunnell, D.J. Hospital presenting self-harm and risk of fatal and non-fatal repetition: systematic review and meta-analysis. PLoS One 2014, 9, e89944, doi:10.1371/journal.pone.0089944.
  • Cipriano, A.; Cella, S.; Cotrufo, P. Nonsuicidal Self-injury: A Systematic Review. Frontiers in psychology 2017, 8, 1946, 643 doi:10.3389/fpsyg.2017.01946.
  • Bardone-Cone, A.M.; Thompson, K.A.; Miller, A.J. The self and eating disorders. Journal of Personality 2020, 88, 59-75, doi:https://doi.org/10.1111/jopy.12448.
  • Arcelus, J.; Haslam, M.; Farrow, C.; Meyer, C. The role of interpersonal functioning in the maintenance of eating psychopathology: a systematic review and testable model. Clinical psychology review 2013, 33, 156-167, doi:10.1016/j.cpr.2012.10.009.
  • Smink, F.R.; van Hoeken, D.; Hoek, H.W. Epidemiology of eating disorders: incidence, prevalence and mortality rates. Current psychiatry reports 2012, 14, 406-414, doi:10.1007/s11920-012-0282-y.
  • Hoek, H.W.; van Hoeken, D. Review of the prevalence and incidence of eating disorders. The International journal of eating disorders 2003, 34, 383-396, doi:10.1002/eat.10222.
  • Van Hoeken, D.; Hoek, H.W. Review of the burden of eating disorders: mortality, disability, costs, quality of life, and family burden. Current opinion in psychiatry 2020, 33, 521-527, doi:10.1097/yco.0000000000000641.

The following reviews are referenced in the introduction section to indicate that the functions of SH have been explored to a greater extent than ED behaviours. To the best of the authors knowledge a systematic review has not been completed specifically exploring the functions of ED behaviours. 

  • Klonsky, E.D. The functions of deliberate self-injury: a review of the evidence. Clinical psychology review 2007, 27, 226-239, doi:10.1016/j.cpr.2006.08.002.
  • Suyemoto, K.L. The functions of self-mutilation. Clinical psychology review 1998, 18, 531-554, doi:10.1016/s0272-7358(97)00105-0.
  • Edmondson, A.J.; Brennan, C.A.; House, A.O. Non-suicidal reasons for self-harm: A systematic review of self-reported accounts. Journal of Affective Disorders 2016, 191, 109-117, doi:https://doi.org/10.1016/j.jad.2015.11.043.

In addition, the following report summarising advances in the relationship between ED behaviours and SH and is included in the introduction:

  • Kiekens, G.; Claes, L. Non-Suicidal Self-Injury and Eating Disordered Behaviors: An Update on What We Do and Do Not Know. Current psychiatry reports 2020, 22, 68, doi:10.1007/s11920-020-01191-y.

We would very much welcome any feedback on specific systematic reviews that we are missing which would help to strengthen the introduction section.

Comment: PLEASE RECORD THE PROTOCOL FOR THIS PAPER. THIS PAPER CONCERNS, IN ADDITION TO EATING DISORDERS, THEIR RELATIONSHIP WITH SELF-HARMING BEHAVIORS. ​​The protocol indicated corresponds to previous work.

Reply: Thank you for flagging that the title of the PROSPERO registration differs from the title of the submitted manuscript. This record refers to the same review as the manuscript, however, the work had evolved iteratively. The proposed search strategy, eligibility criteria, outcomes and data collection/synthesis of the PROSPERO record all remain aligned with the current manuscript. Please see below aims recorded on the PROSPERO record. As the review progressed the additional outcome became more of a focus and therefore the title was adapted to reflect this.

Main outcomes

The proposed qualitative evidence synthesis aims to identify and summarise published first-hand qualitative accounts of the functions of Eating Disorder behaviours (restrictive eating, binge eating, and compensatory behaviours such as self-induced vomiting, misuse of laxatives and excessive exercise) expressed by individuals who experience eating disorder behaviours.

Additional outcomes

A secondary outcome, where possible, will be to compare the similarities and differences in functions between eating disorder behaviours and self-harm.

Comment: If possible, include the search algorithms used for the remaining databases in the Supplementary Material – Search Strategy. Verify that they work correctly.

Reply: The authors kept a detailed record of each search strategy as well as the results of each search. We have noted in the manuscript that search algorithms for the databases other than EMBASE are available upon request.

Reviewer 4 Report

Comments and Suggestions for Authors

Dear Authors,

Thank you for providing me with the opportunity to review this interesting piece of paper. Below, I have listed my comments:

1) In the introduction, a clearer definition or explanation of Restrictive Intake Self-Harm (RISH) would benefit readers unfamiliar with this emerging term.

2) In the selection criteria, the exclusion of orthorexia and emotional eating is noted, but why these were excluded isn’t well-justified. Since emotional eating may also have coping/functional roles, this decision could be briefly explained.

3) The presentation of the results is a little complex, making it hard for the reader to follow it. Some themes (e.g., validation, interpersonal influence, protection) overlap in conceptual scope and could benefit from clearer boundaries or cross-referencing. Quotes that illustrate the same idea (e.g., using ED for control or comfort) are often too numerous. It is best to select fewer, more representative ones to avoid saturation.

4) Although 21 countries are included, cultural or regional influences on ED behaviour are not deeply discussed. a brief commentary on how cultural context (e.g., Western beauty norms, collectivist vs. individualist societies) might shape different functions or expressions of ED behaviour might be of help.

5) Themes like to die”, “responding to body ideals”, or “delaying adulthood” are labeled as unique to EDs, but it's acknowledged that some also appear in SH contexts. Perhaps, instead of labeling them strictly “unique,” consider framing them as “emergent themes more salient in ED literature” to acknowledge nuance.

6) Regarding the paper as a whole, there is no discussion of the ethical considerations involved in interpreting distressing quotations or presenting suicide-related motives. One suggestion I would make is to add a few lines—either within the Discussion section or another area you feel is more appropriate—explaining how such data were handled sensitively, and what this means for clinical practice and future research ethics.

I hope this feedback is helpful.

Author Response

Dear Reviewer 4,

Thank you very much for taking the time to review this manuscript. Please see below a point-by-point response to your comments in red text. Corresponding revisions/corrections can be seen in tracked changes in the re-submitted files.

Reviewer comments:

Dear Authors,

Thank you for providing me with the opportunity to review this interesting piece of paper. Below, I have listed my comments:

1) In the introduction, a clearer definition or explanation of Restrictive Intake Self-Harm (RISH) would benefit readers unfamiliar with this emerging term.

Reply: Thank you very much for this suggestion, we agree that a clearer definition of RISH is needed and have therefore added the following definition into the manuscript:

“a formulation-driven term which aims to describe the subset of patients who present with restricted intake (both foods and fluids) as a method of indirect self-harm”

2) In the selection criteria, the exclusion of orthorexia and emotional eating is noted, but why these were excluded isn’t well-justified. Since emotional eating may also have coping/functional roles, this decision could be briefly explained.

Reply: We completely agree with regards to justifying this decision. In line with reviewer 2 comments, we have amended the way we have presented the inclusion criteria. We hope that the following rephrasing helps to better explain the decision to exclude orthorexia and emotional eating:

“The decision was made to include only recognised ED behaviours according to the ICD 10 (International Statistical Classification of Diseases and Related Health Problems 10th Revision) criteria. As a result, behaviours such as orthorexia and emotional eating, which are not recognised within this classification system were excluded.”

3) The presentation of the results is a little complex, making it hard for the reader to follow it. Some themes (e.g., validation, interpersonal influence, protection) overlap in conceptual scope and could benefit from clearer boundaries or cross-referencing. Quotes that illustrate the same idea (e.g., using ED for control or comfort) are often too numerous. It is best to select fewer, more representative ones to avoid saturation.

Reply: Thank you for your thoughtful feedback regarding the results section. We agree that the results are complex and have therefore (in line with reviewer 1 comments) added a conceptual figure summarising the overlap vs distinctions in functions between SH and ED. We hope this will support readers in following the results section. We have also removed some quotations, as suggested.

4) Although 21 countries are included, cultural or regional influences on ED behaviour are not deeply discussed. a brief commentary on how cultural context (e.g., Western beauty norms, collectivist vs. individualist societies) might shape different functions or expressions of ED behaviour might be of help.

Reply: The potential influence of culture/region on ED behaviours is a very important consideration although something we felt we could not comprehensively discuss within this review. Therefore, we have acknowledged this lack of discussion as a limitation:

“There is the potential for culture/region to influence functionalities of ED behaviours across different populations, however comprehensively exploring this was something that was outside of the scope of this review. Therefore, future research that explores how cultural and contextual factors influence the functionality of ED behaviours may be beneficial.”

5) Themes like “to die”, “responding to body ideals”, or “delaying adulthood” are labelled as unique to EDs, but it's acknowledged that some also appear in SH contexts. Perhaps, instead of labelling them strictly “unique,” consider framing them as “emergent themes more salient in ED literature” to acknowledge nuance.

Reply: Thank you for this valuable suggestion. On reflection, we agree that labelling something as strictly unique does not acknowledge potential nuance. Therefore, as suggested we have rephrased “unique ED functions” to “emergent functions more salient in the ED literature.” This has also been done for SH, replacing “unique SH functions” with “Functions more salient in the SH literature.”

6) Regarding the paper as a whole, there is no discussion of the ethical considerations involved in interpreting distressing quotations or presenting suicide-related motives. One suggestion I would make is to add a few lines—either within the Discussion section or another area you feel is more appropriate—explaining how such data were handled sensitively, and what this means for clinical practice and future research ethics.

Reply: This is a really interesting point and in retrospect, potential ethical issues are something that should have been addressed in the original manuscript. Therefore, the following section has been added:

2.6 Ethical Considerations

“Whilst no new empirical data was gathered for the purpose of this review, the team were still mindful of potential ethical considerations and maintaining respect for the data provided by original participants. The review team are experienced in managing sensitive data and the dataset was not shared outside of the review team. With regards to informed consent, quotations were extracted from published studies in which participants had consented for their quotations to be used publicly. Only anonymised data was available via included studies and therefore the risk of individual identification of participants at review stage was minimal. Quality assessment processes also took account of ethical considerations within each individual study.

The sensitive nature of this topic and the potential for distress was frequently considered during the review process. The risk of researcher distress was mitigated through supervision and reflective discussions. Sensitive presentation of this research was a key consideration for the team, particularly around the use of language.

Round 2

Reviewer 1 Report

Comments and Suggestions for Authors

Dear Authors,

Thank you for your detailed responses and the considerable effort you have put into revising the manuscript. The additional methodological details, particularly regarding data management, inter-coder reliability, and iterative theme refinement, are clear and strengthen the transparency of the study.

The integration of transdiagnostic frameworks and gender schema theory enriches the discussion, and the conceptual figure is a valuable addition to illustrate shared and distinct functions of ED and SH behaviours. Terminology is now consistent, and the abstract aligns well with reporting standards.

I am satisfied that my previous comments have been fully addressed, and I believe the manuscript has improved significantly in clarity and rigour. I have no further substantial comments, and I consider it suitable for publication after the usual editorial checks.

Best regards,

Reviewer 4 Report

Comments and Suggestions for Authors

Thank you for revising the paper. Good luck with the rest of the process.